# Translational control of auditory imprinting and structural plasticity by eIF2α

**Gervasio Batista[1]\*, Jennifer Leigh Johnson[2], Elena Dominguez[1], Mauro Costa-Mattioli[2], Jose L Pena[1]\***

[1]Dominick P. Purpura Department of Neuroscience, Albert Einstein College of Medicine, New York, United States; [2]Department of Neuroscience, Baylor College of Medicine, Houston, United States

**Abstract** The formation of imprinted memories during a critical period is crucial for vital behaviors, including filial attachment. Yet, little is known about the underlying molecular mechanisms. Using a combination of behavior, pharmacology, in vivo surface sensing of translation (SUnSET) and DiOlistic labeling we found that, translational control by the eukaryotic translation initiation factor 2 alpha (eIF2α) bidirectionally regulates auditory but not visual imprinting and related changes in structural plasticity in chickens. Increasing phosphorylation of eIF2α (p-eIF2α) reduces translation rates and spine plasticity, and selectively impairs auditory imprinting. By contrast, inhibition of an eIF2α kinase or blocking the translational program controlled by p-eIF2α enhances auditory imprinting. Importantly, these manipulations are able to reopen the critical period. Thus, we have identified a translational control mechanism that selectively underlies auditory imprinting. Restoring translational control of eIF2α holds the promise to rejuvenate adult brain plasticity and restore learning and memory in a variety of cognitive disorders.

**\*For correspondence:** gervasio. batista@phd.einstein.yu.edu (GB); jose.pena@einstein.yu.edu (JLP)

**Competing interests:** The authors declare that no competing interests exist.

## Introduction

Imprinting is a form of early learning where exposure to a stimulus becomes the triggering signal of a vital behavior (*Jin et al., 2016*; *Horn, 2004*). A particular feature of imprinting is that it occurs exclusively within a short critical period (CP) (*Jin et al., 2016*; *Bolhuis, 1991*; *Nevitt et al., 1994*), when structural and functional changes take place (*Hensch, 2004*). Imprinting drives a vigorous following behavior in chickens, key for filial attachment (*Horn, 2004*; *Insel and Young, 2001*). This rather unique and precocious behavior is advantageous for investigating experience-driven activation of molecular pathways around birth (*Bredenkötter and Braun, 1997*; *Bock and Braun, 1999*; *McCabe et al., 1982*, *1981*). Understanding the biological basis of imprinting can shed light on the mechanisms of learning in newborns and create new avenues to rejuvenate adult brain plasticity by reopening CPs.

The formation of imprinted memories has been described across sensory modalities (*Nevitt et al., 1994*; *McCabe et al., 1982*; *Remy and Hobert, 2005*; *Bock et al., 1997*). Interestingly, in chickens, auditory and visual imprinting relies on different brain structures. Imprinted sounds activate the mediorostral nidopallium/mesopallium (MNM) (*Bock and Braun, 1999*; *Wallhäusser and Scheich, 1987*), where neural responsiveness increases after training (*Bredenkötter and Braun, 1997*). In contrast, the intermediate medial mesopallium (IMM, former IMHV) (*Horn, 2004*; *McCabe et al., 1982*) is required for visual imprinting, where neural responses shift to favor the imprinted object (*Horn et al., 2001*). While the brain circuits and neurophysiological changes have been uncovered (*Horn, 2004*; *Scheich, 1987*), much less is known about the

**eLife digest** Shortly after hatching, a chick recognizes the sight and sound of its mother and follows her around. This requires a type of learning called imprinting, which only occurs during a short period of time in young life known as the "critical period". This process has been reported in a variety of birds and other animals where long-term memory formed during a critical period guides vital behaviors. In order to form imprinted memories, neurons must produce new proteins. However, it is not clear how new experiences trigger the production of these proteins during imprinting. Unraveling such mechanisms may help us to develop drugs that can recover plasticity in the adult brain, which could help individuals with brain injuries relearn skills after critical periods are closed.

It is possible to imprint newly hatched chicks to arbitrary sounds and visual stimuli by placing the chicks in running wheels and exposing them to repeated noises and videos. Later on, the chicks respond to these stimuli by running towards the screen, mimicking how they would naturally follow their mother. This system allows researchers to measure imprinting in a carefully controlled laboratory setting.

A protein called eIF2α plays a major role in regulating the production of new proteins and has been shown to be required for the formation of long-term memories in adult rodents. Batista et al. found that eIF2α is required to imprint newly hatched chicks to sound. During the critical period, this factor mediates an increase in "memory-spines", which are small bumps on neurons that are thought to be involved in memory storage. On the other hand, eIF2α was not required to imprint newly hatched chicks to visual stimuli, suggesting that there are different pathways involved in regulating imprinting to different senses. Batista et al. also demonstrate that using drugs to increase the activity of eIF2α in older chicks could allow these chicks to be imprinted to new sounds.

The next steps following on from this work are to identify proteins that eIF2α regulates to form memories, and to find out why eIF2α is only required to imprint sounds. Future research will investigate the mechanisms that control visual imprinting and how it differs from imprinting to sounds.

molecular machinery linking experience and the formation of imprinted memories in each sensory modality.

While imprinting requires protein synthesis (*Gibbs and Lecanuet, 1981*), little is known about the underlying translational control mechanisms. The translation of mRNA into protein occurs in three steps: initiation, elongation and termination and can be regulated through several signaling pathways (*Sonenberg and Hinnebusch, 2009*). Translation initiation is believed to be the rate-limiting step and a key target for translational control (*Sonenberg and Hinnebusch, 2009*; *Buffington et al., 2014*). A major way in which translation initiation is regulated is by modulating the formation of the ternary complex *via* phosphorylation of the translation-initiation factor eIF2α. In rodents, protein synthesis controlled by phosphorylation of eIF2α is critically required for long-lasting forms of synaptic plasticity (*Costa-Mattioli et al., 2007*; *Di Prisco et al., 2014*) as well as long-term memory storage in several systems (*Costa-Mattioli et al., 2007*, *2005*; *Zhu et al., 2011*; *Stern et al., 2013*; *Ounallah-Saad et al., 2014*; *Ma et al., 2013*). Here we asked whether this central translational control mechanism plays a role in imprinting in newborn chickens and can be used to restore imprinting outside of the CP.

## Results

### Critical periods for visual and auditory imprinting

Dark reared chickens were placed in a running wheel in front of an LCD screen and a speaker for training. Visual and auditory imprinting were tested separately 24 hr after training (*Figure 1a*). Stimuli consisted of animated movies showing a virtual object, and artificial sounds synchronized to movements of the object in the screen (*Figure 1b*, see supplementary materials). The imprinting was assessed by the preferential approach to the imprinted stimuli, either visual or auditory, compared to the approach to novel stimuli. The preference for imprinted stimuli is commonly used as an index

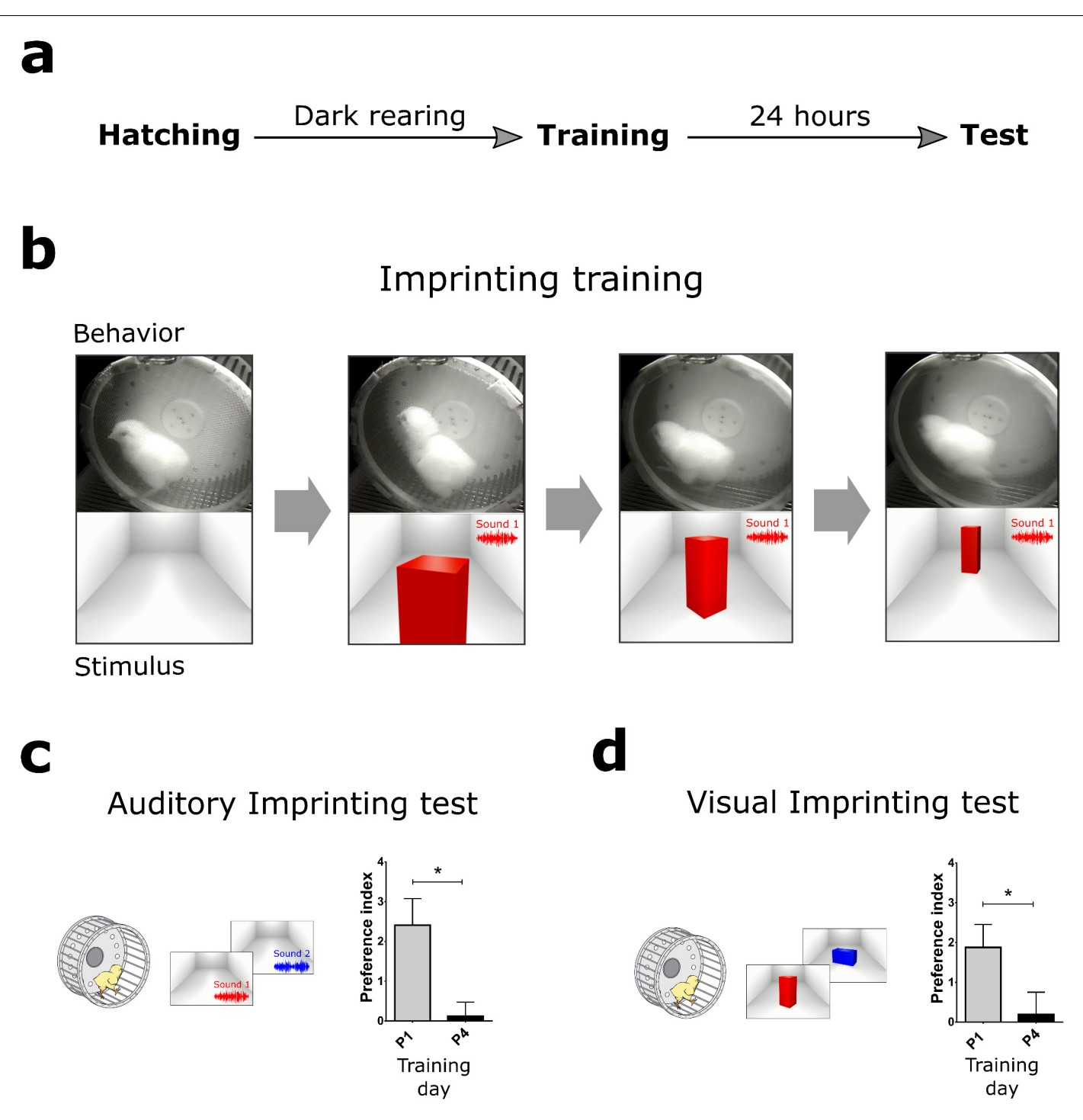

**Figure 1.** Behavioral paradigm and the critical period for imprinting. (**a**) Schematic sequence of behavioral experiments. Dark-reared chicks were trained in a running wheel and tested the day after for visual and auditory imprinting. (**b**) During imprinting training, the chickens were presented with audiovisual stimulation. An animated object moved across the screen while a sound was presented every 3 s, coupled with pulsating movements of the object. (**c**) Auditory imprinting (left) was assessed by comparing the approaching behavior on the wheel to the imprinted sound or a novel sound. This procedure generated robust auditory imprinting when training was performed the day after hatching (gray, n = 13) but was ineffective four days after hatching (black, n = 12) (right). (**d**) Visual imprinting (left) was assessed independently, by comparing the approaching behavior to the imprinted or a novel image. Similarly to auditory imprinting, visual imprinting was strong in P1 (gray, n = 13) but absent in P4 (black, n = 12) (right). Plots show mean and SEM, * indicates p<0.05 from two-sample t-test.

*Figure 1 continued on next page*

*Figure 1 continued*

The following source data is available for figure 1:

**Source data 1.** Preference indexes of trained chickens during (P1) or after the critical period (P4).

of long-term memory storage (*Horn, 2004*). Individuals' preference was measured by calculating an index, where positive and negative values indicate preference for the imprinted or novel stimulus, respectively (*Figure 1c and d*). This index accounts for fluctuations in baseline locomotion across trials, as described in the Method section. Consistent with previous studies (*Yamaguchi et al., 2012*), chickens showed imprinting to either visual or auditory cues one day after hatching (P1) but not after four days (P4) (*Figure 1c and d*), indicating that the CP for imprinting ends before P4.

## Protein-synthesis dependency of auditory and visual imprinting

To assess whether a protein synthesis is enhanced after imprinting in both MNM and IMM we optimized an in vivo surface sensing of translation (SUnSET) protocol (*Schmidt et al., 2009*) for monitoring protein synthesis in vivo in these areas. Briefly, the antibiotic puromycin (PMY) incorporated into newly synthesized proteins can be detected through immunolabeling and used to monitor translation. Because brain tissue incorporates PMY more slowly compared to other tissues (*Flexner et al., 1962*) pilot experiments were conducted, showing that IP-injected PMY accessed the chicken's brain within 3–4 hr. Thus PMY was injected 1 hr before a 2 hr training and samples were collected 4 hr after injection to capture training-induced translation.

We found that imprinting training increased translation both in MNM and IMM (*Figure 2b,c*) after a 2 hr training session, compared to controls, which were running on the wheel but presented with an empty screen. To further estimate the time-window during which auditory and visual imprinting are sensitive to protein synthesis inhibition, we trained chickens for 1 or 2 hr on P1. Two-hour (*Figure 3a*, right panel) training triggered robust auditory imprinting, which was blocked by the protein synthesis inhibitor cycloheximide (CHX) injected immediately after training (*Figure 3a*, left panel). In contrast, one-hour training did not elicit significant auditory imprinting (*Figure 3a*, right panel). Interestingly, the temporal dynamics of protein synthesis dependency of visual imprinting was different. While one-hour training triggered visual imprinting that was suppressed by CHX (*Figure 3b*, left panel), visual imprinting after two-hour training was not blocked by post-training administration of CHX (*Figure 3b* right panel). Consistent with the effect on behavior, CHX effectively blocked imprinting-triggered protein synthesis in both MNM and IMM areas (*Figure 3c,d*). Taken together, our results show that both auditory and visual imprinting trigger new protein synthesis, which is required for both auditory and visual imprinting.

## eIF2α-mediated translational control selectively regulates auditory imprinting

To investigate whether the translational program controlled by eIF2α is involved in imprinting, we first measured levels of phosphorylated eIF2α (p-eIF2α) in MNM and IMM after training P1 chickens. Intriguingly, training significantly decreased p-eIF2α in the auditory area MNM (*Figure 4a*, left panel), but not in the visual area IMM (*Figure 4a*, right panel). To examine whether a reduction in eIF2α phosphorylation is required for auditory imprinting we treated chickens before training with Sal003, an inhibitor of the eIF2α phosphatase complexes (*McCamphill et al., 2015*), which increases p-eIF2α levels (*Figure 4b* and *Figure 4—figure supplement 1*) and decreases translation (*Figure 4—figure supplement 2*). Interestingly, increasing p-eIF2α with Sal003 prevented auditory imprinting, but had no effect on visual imprinting (*Figure 4c*). These results indicate that decreasing p-eIF2α is only required for auditory imprinting.

We next asked whether decreasing p-eIF2α would selectively enhance auditory imprinting. To this end, we first blocked the activity of the eIF2α kinase PKR, with a specific PKR inhibitor (*Zhu et al., 2011*) (PKRi). PKRi-injected chickens showed significantly stronger auditory imprinting compared to controls (*Figure 4d*). However, PKRi failed to affect visual imprinting (*Figure 4d*). Given that average locomotion towards the computer screen in both treated and control conditions was similar, the changes induced by altering eIF2α phosphorylation cannot be attributed to changes in

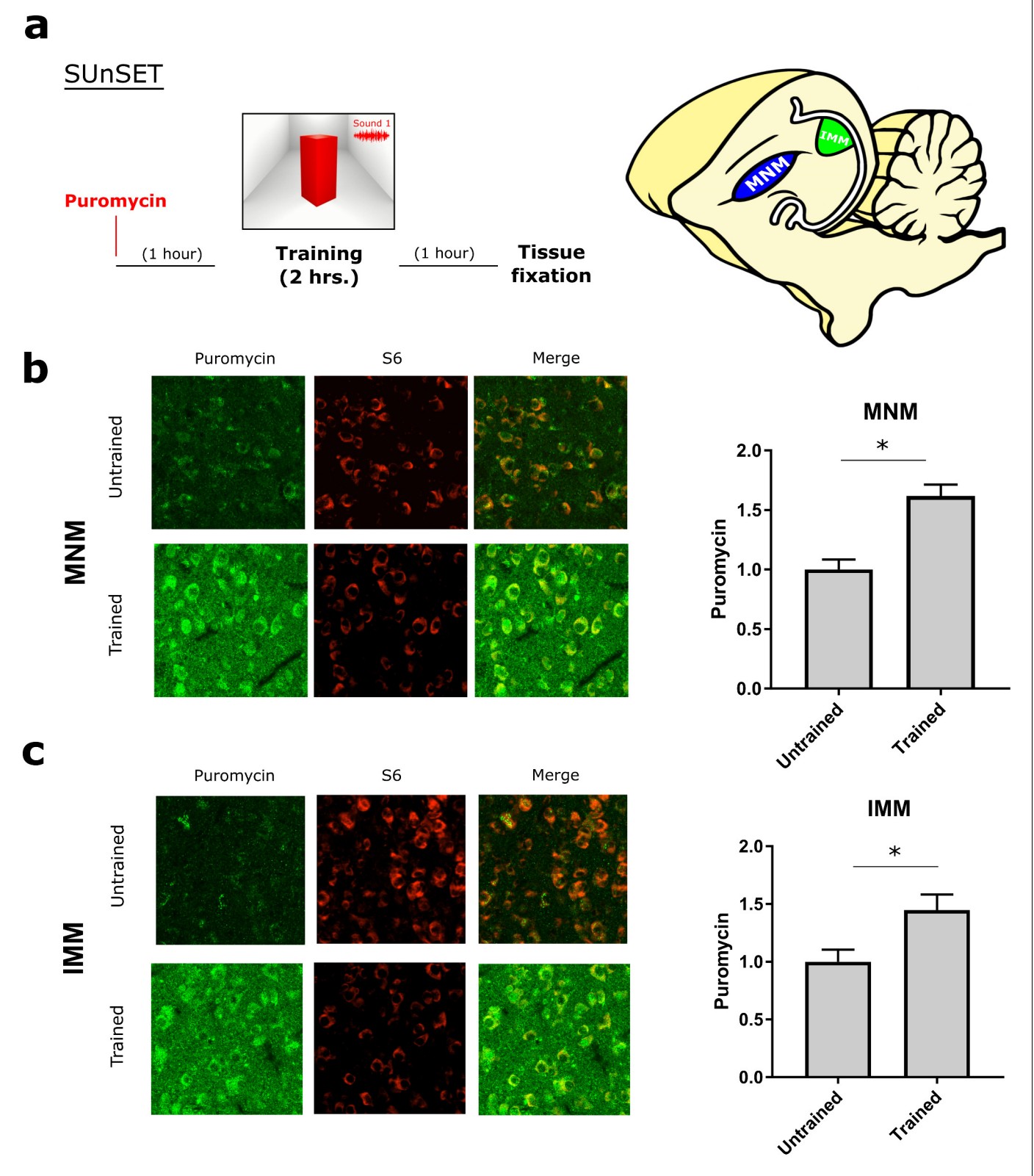

**Figure 2.** Experience-dependent increase in translation assessed with SUnSET. (a) Temporally optimized SUnSET protocol to detect changes in translation in vivo (left) induced by the imprinting training. Schematic sagittal view of the chicken forebrain, showing the position of MNM and IMM (right). (b) The auditory imprinting area MNM (left) exhibits increased puromycin incorporation (green) after imprinting training, compared with MNM samples of chickens running on the wheel but not presented with the imprinting object. S6 (red) marker was used to identify cells somas. (c) In IMM

*Figure 2 continued on next page*

*Figure 2 continued*

(left) translation rates were also increased in trained animals. Sample sizes: MNM untrained (six chickens, 48 images at 10X, zoom 3X); MNM trained (six chickens, 48 images at 10X, zoom 3X); IMM untrained (six chickens, 48 images at 10X, zoom 3X); IMM trained (six chickens, 48 images at 10X, zoom 3X). Bar plots show mean and SEM; * indicates p<0.05 from unpaired Mann-Whitney test.

The following source data is available for figure 2:

**Source data 1.** SUnSET results from trained and untrained chickens.

overall motor activity. To further demonstrate that auditory imprinting could be enhanced by reducing eIF2α-mediated translational control, chickens were injected with ISRIB, a compound that blocks the translational effects induced by p-eIF2α (*Sidrauski et al., 2013*) and increases translation (*Figure 4—figure supplement 2*). Consistent with the PKRi-experiments, injection of ISRIB immediately after training enhanced auditory imprinting (*Figure 4e*) but not visual imprinting. Hence, a reduction in p-eIF2α-mediated translational control enhances auditory but not visual imprinting.

## eIF2α dephosphorylation is required only for experience-dependent structural plasticity in the auditory imprinting pathway

Plasticity in dendritic spines, the major site of excitatory inputs in neurons, is thought to be crucial during CPs (*Roberts et al., 2010*) and part of the cellular substrate of memory (*Lamprecht and LeDoux, 2004*; *Bourne and Harris, 2007*; *Nishiyama and Yasuda, 2015*). Given that (a) long-term remodeling of spines requires protein synthesis (*Nishiyama and Yasuda, 2015*) and (b) translational control by p-eIF2α selectively regulates auditory imprinting, we next examined the role of this translational control mechanism in structural plasticity in imprinting-relevant brain regions. To measure changes in dendritic spine number and morphology after training (*Figure 5b*), we used the sparse Diolistic labeling technique (*Figure 5a*). The spines were classified (by observers blind to treatment) in stubby, filopodia, thin and mushroom (*Figure 5c*), a method that is informative about the functionality and maturity of spines (*Bourne and Harris, 2007*) and has been used in studies of learning-related structural plasticity (*Sanders et al., 2012*).

While training failed to affect the total number of spines (*Figure 5d–e*), it significantly increased the number of mushroom spines and decreased the number of thin spines in MNM (*Figure 5d*) and IMM (*Figure 5e*), compared to control animals with experience on the running wheel but not subject to audiovisual training. We next examined whether blocking eIF2α dephosphorization with Sal003 prevents training-induced changes in structural plasticity. Remarkably, Sal003 administration blocked the training-induced increase in the number of mushroom spines only in MNM (*Figure 5d,e*). These results indicate that eIF2α phosphorylation not only controls the imprinting behavior during the CP but also structural plasticity, a potential cellular substrate of memory storage (*Lamprecht and LeDoux, 2004*; *Nishiyama and Yasuda, 2015*) in a key forebrain area involved in auditory imprinting.

## Blocking p-eIF2α mediated translation reopens the critical period for auditory imprinting

Identifying the mechanisms that open the CPs could lead to novel therapeutic opportunities for a variety of cognitive disorders (*Hensch, 2004*). Given that (a) behavioral training decreases p-eIF2α (*Figure 4a*), (b) blocking p-eIF2α-mediated translation enhances auditory imprinting elicited by weak-training protocol (*Figure 4d–e*) and (c) Sal003-mediated increase in p-eIF2α blocks auditory imprinting, we wondered whether the drugs enhancing imprinting during the CP (PKRi and ISRIB) would restore imprinting outside the CP (*Figure 6a*). Remarkably, treatment with either PKRi or ISRIB (*Figure 6a*) on P4 selectively re-opened the CP for auditory imprinting (*Figure 6b*), again without affecting visual imprinting (*Figure 6c*). Hence, by promoting brain plasticity, the reduction of p-eIF2α-mediated translational control enhances auditory imprinting.

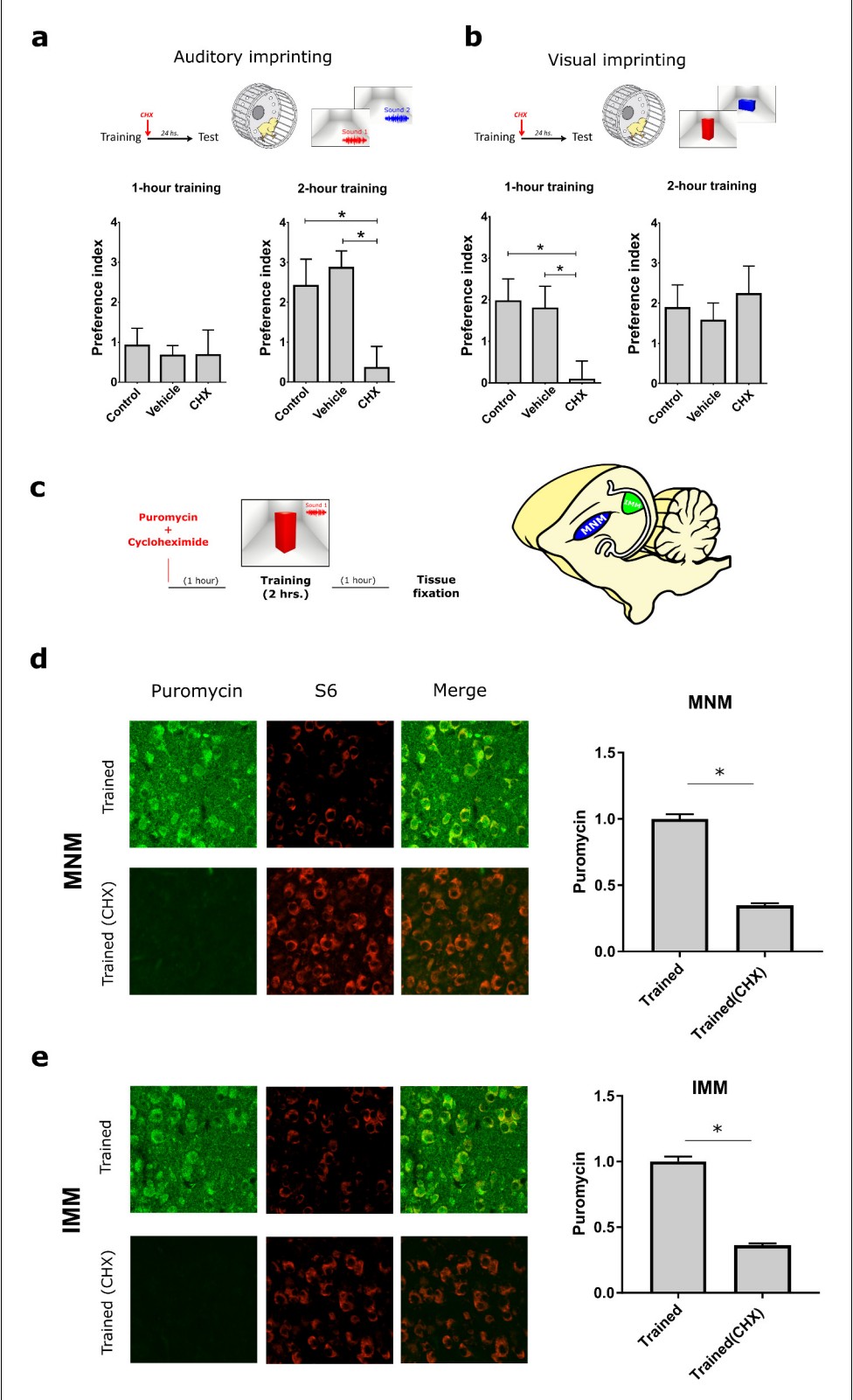

**Figure 3.** Protein synthesis requirement in auditory and visual imprinting. (a) The protein-synthesis inhibitor cycloheximide (CHX, n = 12) injected immediately after 1 hr training (left) had no effect on the auditory preference index compared to controls (n = 15) and vehicle-injected (n = 14) groups. In contrast, 2 hr training, which induced stronger preference to the imprinted sound, was blocked by CHX-treatment (n = 11) compared to controls (n = 13) and vehicle-injected group (n = 9). (b) Visual imprinting was already robust after 1 hr training (left) in controls (n = 15) and chickens injected with

*Figure 3 continued on next page*

*Figure 3 continued*

vehicle (n = 14), and blocked by CHX-administration (n = 12). On the other hand, 2 hr training (right) also induced robust preference to the imprinted visual object in controls (n = 13) and vehicle-injected chickens (n = 9) but was not blocked by CHX administration (n = 12). Plots show mean and SEM, * indicates p<0.05 from two-ways ANOVA test, Bonferroni *Post hoc* test. (c) SUnSET protocol used to detect experience-dependent translation changes in MNM and IMM in the presence or absence of CHX. (d,e) Puromycin (green) incorporation is decreased in trained animals treated with CHX. S6 (red) was used to identify cell somas. Sample sizes: MNM trained (five chickens, 40 images at 10X, zoom 3X); MNM trained and CHX administration (six chickens, 48 images at 10X, zoom 3X); IMM trained (five chickens, 39 images at 10X, zoom 3X); IMM trained and CHX administration (six chickens, 47 images at 10X, zoom 3X). Bar plots show mean and SEM; * indicates p<0.05 from unpaired Mann-Whitney test.

The following source data is available for figure 3:

**Source data 1.** Preference indexes and SUnSET results from control chickens and injected with cycloheximide.

## Discussion

### Regulation of imprinting and structural plasticity by eIF2α

Imprinting allows newborns to adjust behavior in response to relevant sensory experience, immediately after birth (*Horn, 2004*; *Bolhuis, 1991*). Despite having been studied for decades, the mechanism mediating the formation of imprinted memories remains elusive. Here we showed that, although visual and auditory imprinting require newly synthesized proteins, eIF2α-mediated translational control bidirectionally regulates auditory but not visual imprinting and related changes in structural plasticity. Remarkably, targeting this translational control mechanism pharmacologically recovers auditory imprinting after the closing of the critical period.

Critical periods in the auditory system have been widely studied across species (*Scheich, 1987*; *Riebel et al., 2002*; *Yang et al., 2012*; *Insanally et al., 2009*). Yet the mechanisms engaged during the CP for auditory imprinting have not been elucidated. One major limitation has been the design of stringent experimental approaches that control for social experience and innate biases, while achieving robust auditory imprinted memories (*Van and Bolhuis, 1991*). We aimed to address these concerns by: (1) raising chickens in darkness and constraining social interaction, (2) imprinting chickens to more than one type of object and sound, and (3) increasing the length of training compared to other studies (*Wallhäusser and Scheich, 1987*; *Van and Bolhuis, 1991*) to achieve significant memory retention longer than 24 hr after training. These improvements, in addition to the novel custom-made audiovisual animation used for training, provided a stronger experimental design for assessing auditory and visual imprinting.

Several lines of evidence support the modality-specific role of eIF2α. First, training decreased phosphorylation of eIF2α in the auditory-imprinting relevant area MNM, but not in IMM (*Figure 4a*). Second, pharmacologically increasing eIF2α phosphorylation with Sal003 selectively disrupted auditory imprinting (*Figure 4c*). Third, inhibiting eIF2α phosphorylation with PKRi or directly blocking p-eIF2α-mediated translational control with ISRIB, enhanced auditory imprinting after weak training (*Figure 4d–e*). Interestingly, although Sal003 and ISRIB altered protein synthesis in IMM, these manipulations had no detectable effect on the formation of visual memories. The reason why eIF2α is not involved in visual imprinting is not yet understood. It is possible that expression of eIF2α kinase or phosphatase complexes differs between visual and auditory areas, or that upstream signaling pathways fail to engage eIF2α dephosphorylation. It would be interesting to test whether other tasks involving memory formation, such as one-trial avoidance learning (*Atkinson et al., 2008*), also require translational control by eIF2α. An appealing idea is that other translational control pathways such those controlled by the mechanistic target of rapamycin mTORC1 (*Sonenberg and Hinnebusch, 2009*) mediate the formation of visual memories.

While previous studies in adult rodents suggest that eIF2α-mediated translation regulates the two major forms of synaptic plasticity (*Costa-Mattioli et al., 2007*; *Zhu et al., 2011*; *Stern et al., 2013*), here we report for the first time that experience-dependent structural plasticity of dendritic spines requires eIF2a dephosphorylation. Furthermore, and consistent with our behavioral results, eIF2α-mediated translation exclusively regulated spine remodeling in the auditory but not in the visual area. This result is particularly important since structural plasticity is crucial during CPs (*Roberts et al., 2010*; *Mataga et al., 2004*) but the underlying molecular mechanisms were

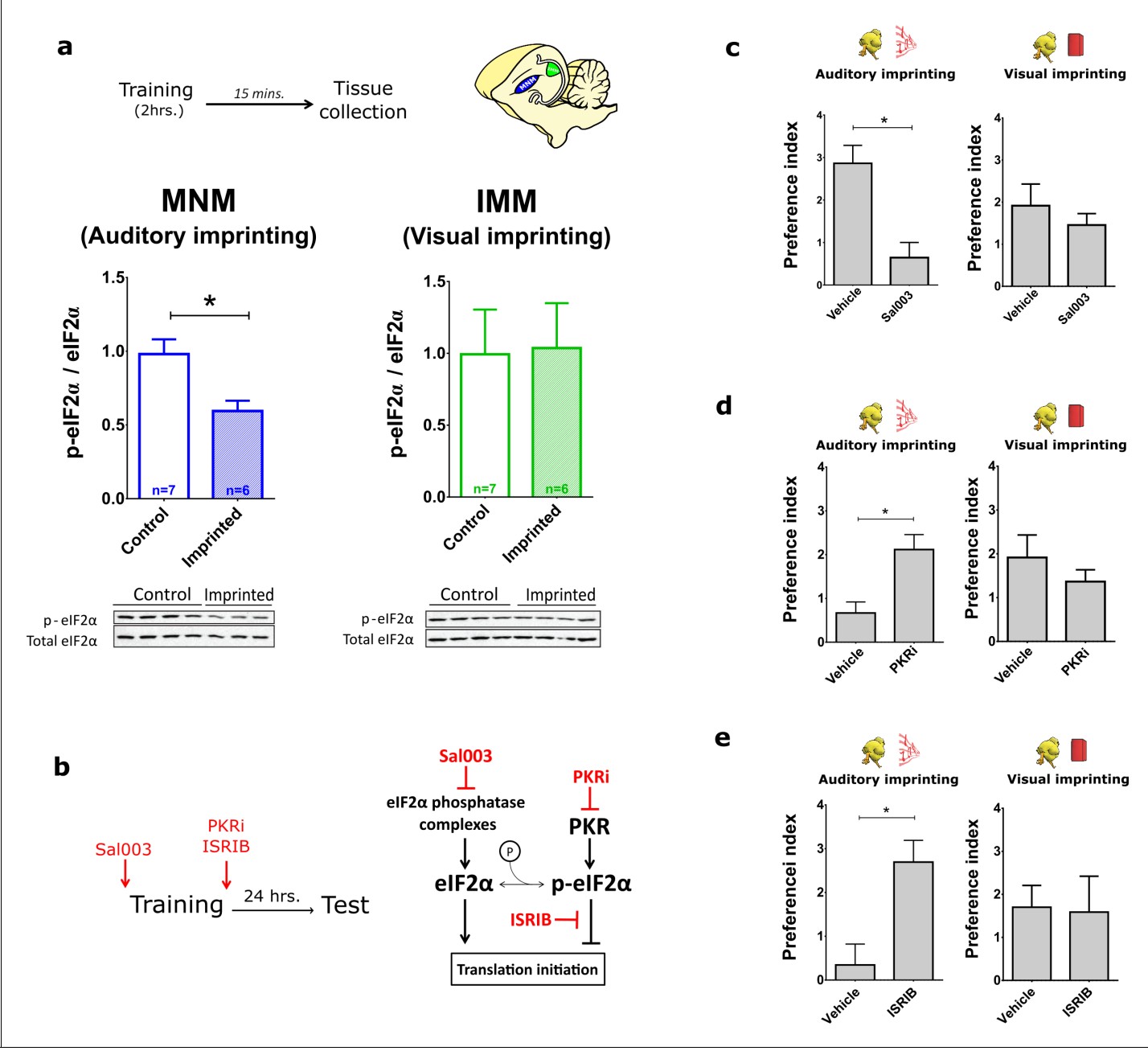

**Figure 4.** Translational control of auditory imprinting by eIF2α. (a) After 2 hr imprinting training, IMM and MNM were punched out for western blot analysis. The ratio of phosphorylated eIF2α (p-eIF2α) and non-phosphorylated eIF2α was measured in controls and after training in MNM (left) and IMM (right) brain tissue. Trained chicks (n = 7) exhibited decreased eIF2α phosphorylation compared to the untrained (n = 6) in MNM but not in IMM. Representative western blots are shown below each panel. * indicates p<0.05 from unpaired Mann-Whitney test. (b) Left, drugs injected for targeting the eIF2α pathway. Right, schematic effect of pharmacological manipulations on the eIF2α pathway. (c) Auditory (left) but not visual (right) imprinting is blocked by Sal003 injection (n = 12) compared to controls injected with vehicle (n = 9). (d) Auditory imprinting (left) was enhanced in chickens injected with the PKR inhibitor PKRi (n = 26), compared to controls injected with saline vehicle (n = 14). On the other hand, PKRi (n = 26) had no effect on visual imprinting (right), compared to saline injection (n = 14). (e) Auditory imprinting (left) but not visual imprinting (right) was enhanced by ISRIB administration (n = 11) compared to controls injected with vehicle (n = 13). Bar plots represent mean and SEM, * indicates p<0.05 from unpaired t-test.

The following source data and figure supplements are available for figure 4:

**Source data 1.** Western blots of p-eIF2α/ total eIF2α ratio and behavioral pharmacology after targeting the eIF2α pathway.

**Figure supplement 1.** Sal003 increases eIF2α phosphorylation.

*Figure 4 continued on next page*

*Figure 4 continued*

**Figure supplement 2.** ISRIB and Sal003 injection modulate translation in vivo.

unknown. Different forms of structural plasticity have been linked to memory, including spine turn-over and morphological changes of preexisting spines (*Lamprecht and LeDoux, 2004*). In this case, the structural plasticity found in IMM and MNM could be consistent with potentiation and enlarge-ment of specific dendritic spines, favoring the detection of imprinted stimuli. While we did not observe changes in spine density, the increase in mushroom spines and decrease in thin spines may suggest coordinated structural plasticity as previously reported in hippocampal slices (*Bourne and Harris, 2011*). Thus, our results shed light on the biological basis of experience-dependent spine remodeling and uncovered eIF2α as a major player in spine remodeling.

Another interesting question is whether translational control by eIF2α in glial cells affects imprint-ing. Glutamate application induces a transient increase of eIF2α phosphorylation in glial cells in vitro (*Flores-Méndez et al., 2013*). This effect has been linked to glutamate removal from the synaptic cleft by glial cells (*Flores-Méndez et al., 2013*). However, its contribution to memory formation in vivo remains untested. In future studies, it will be important to dissect the role of p-eIF2a in memory formation at the cellular level.

### eIF2α-mediated translational control: an evolutionarily conserved mechanism to rejuvenate plasticity and memory?

Behavior is shaped during sensitive periods in early postnatal life, characterized by epochs of height-ened brain plasticity (*Hensch, 2004*; *Nabel and Morishita, 2013*). Reactivating such plasticity in the adult brain has the potential to rehabilitate brain function after CPs are closed (*Hensch, 2004*; *Nabel and Morishita, 2013*; *Hübener and Bonhoeffer, 2014*). This has been successfully achieved in the visual cortex of rodents through direct manipulation of inhibitory synaptic transmission, either pharmacologically (*Hensch et al., 1998*) or through transplantation of embryonic inhibitory neurons (*Davis et al., 2015*). Moreover, in mice and humans, releasing 'epigenetic brakes', could reopen auditory CPs (*Yang et al., 2012*; *Gervain et al., 2013*). Our results uncover a translational control mechanism as a novel target for reopening CPs. Indeed, two different strategies, either blocking p-eIF2α-mediated translation or inhibiting the upstream kinase PKR, enabled chickens to imprint to sounds after the end of the CP, suggesting that blocking p-eIF2α-mediated translation control enhances CP-mediated plasticity. A recent report shows that reducing p-eIF2α-mediated transla-tional control in the VTA can convert adult into adolescent mice with respect to their vulnerability to cocaine-induced changes in synaptic strength and behavior (*Huang et al., 2016*). Based on these results and the evolutionarily conserved nature of this process, we speculate that reopening CPs through blockade of eIF2α-mediated translational control could be used to recover plasticity in the mature brain and treat cognitive dysfunctions.

## Materials and methods

### Animals

We used newly hatched chicks of both sexes from the White Leghorn strain *Gallus gallus domesticus* (Charles River supplier). Fertilized eggs (embryonic ages E14-17) were obtained and subsequently incubated in darkness at 37–38°C under controlled humidity (Grumbach, compact S84). Upon hatch-ing, chickens were transferred to individual compartments of a brooder maintained at 37–38°C (Brin-sea, TLC-5), where they remained in darkness until each experiment. Water and food was provided. It has been shown that chickens are able to eat and drink water in the dark and that this housing does not impact visual acuity or locomotion, compared to chickens reared under light conditions (*Yamaguchi et al., 2012*). These experiments were approved by the institutional animal care com-mittee (IACUC) at Albert Einstein College of Medicine (protocol 20140910).

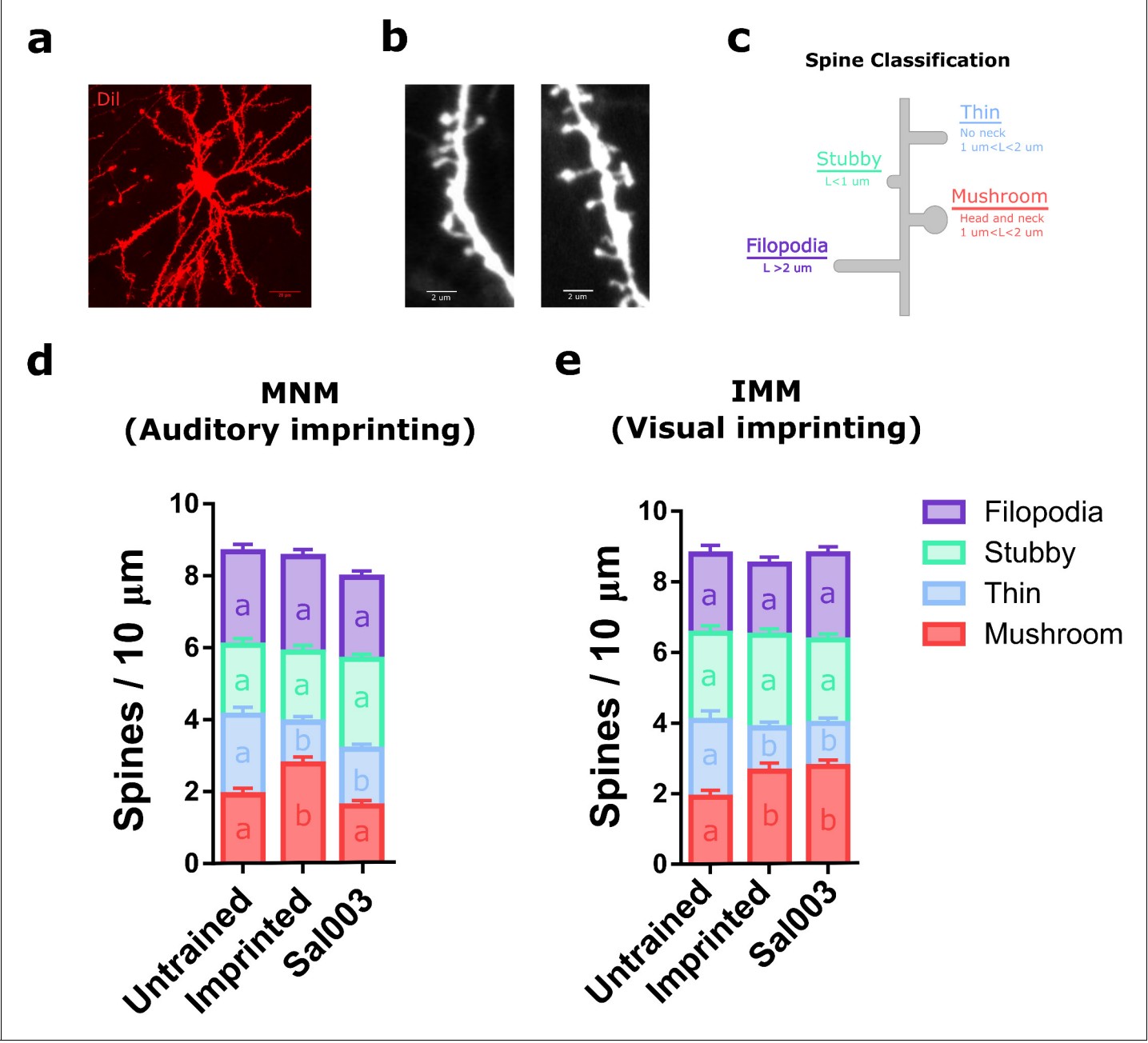

**Figure 5.** Translational control of experience-dependent structural plasticity. (a) Example diolistic labeling of a type I IMM neuron (63X), used to analyze the number and the shape of dendritic spines in MNM and IMM after training. (b) Representative confocal images of dendritic segments of IMM cells from untrained animals (63X, zoom 3X). (c) Schematic length (L) and shape criteria used for spine classification. (d) Trained chickens showed an increased number of mushroom spines (red) and a decrease in thin spines (blue) in MNM. The increase in mushroom spines induced by training was blocked by Sal003. Samples size: untrained (four chickens, 12 cells, 45 dendrites); imprinted (five chickens, 15 cells, 50 dendrites); Sal003 (five chickens, 15 cells, 55 dendrites). (e) Trained chickens showed an increase in mushroom spines (red) and a decrease in thin spines (blue) in IMM. In contrast to the changes in MNM, the increase in mushroom spines was not blocked by Sal003. Sample sizes: untrained (four chickens, 11 cells, 35 dendrites); imprinted (four chickens, 10 cells, 33 dendrites); Sal003 (five chickens, 16 cells, 48 dendrites). Total number of spines did not show significant differences across groups in either region. Bar plots show mean and SEM; different letters inside bars indicate statistically significant differences ($p < 0.05$) between groups from Kruskal-Wallis test, Dunn's multiple comparisons test.

The following source data is available for figure 5:

**Source data 1.** Dendritic spines numbers in MNM and IMM of untrained, trained and Sal003-treated chickens.

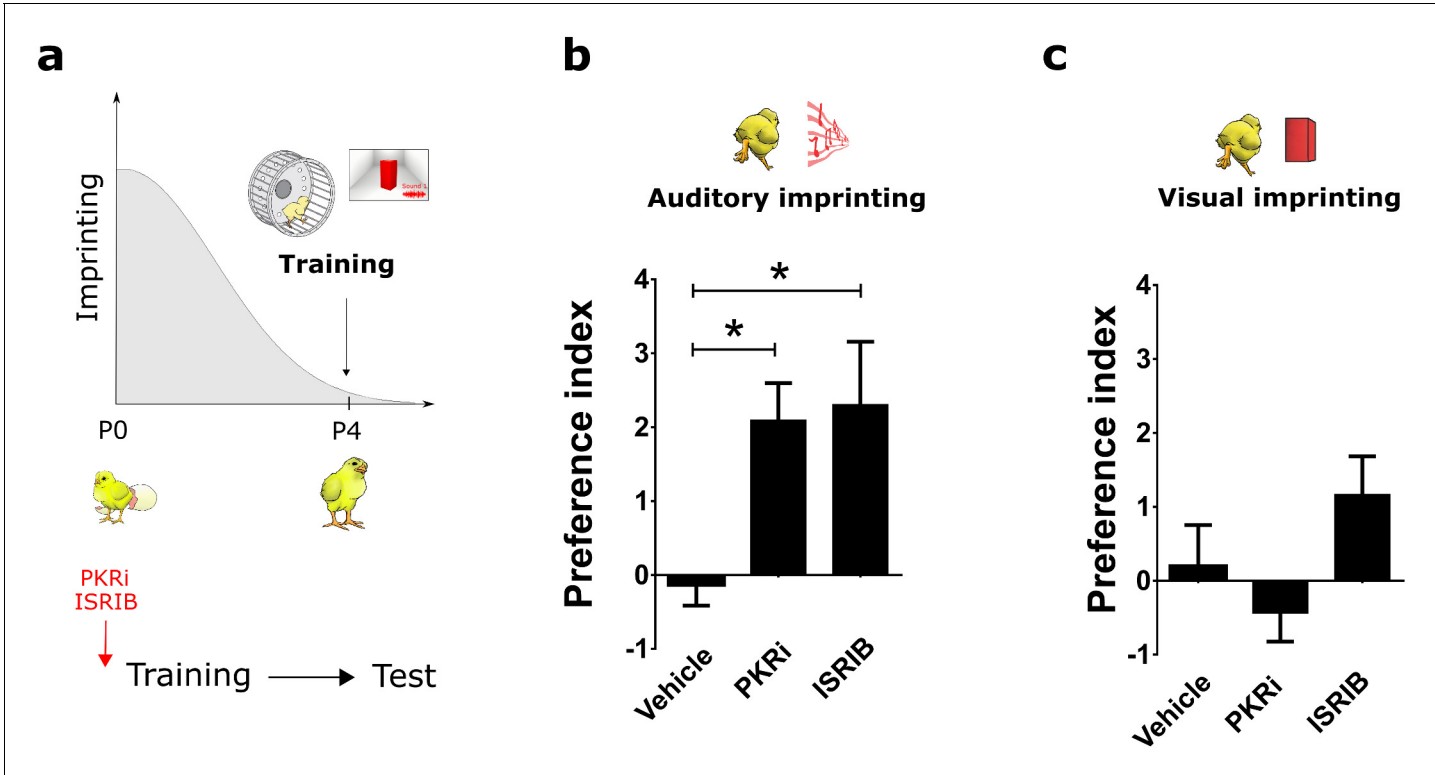

**Figure 6.** Reopening the critical period for visual and auditory imprinting through eIF2α. (a) Chickens were trained 4 days after hatching (P4) and tested 24 hr after training. To target translational control by eIF2α, chickens were injected with PKRi or ISRIB. (b) Controls injected with vehicle (n = 12) did not show auditory imprinting at P4 but the critical period in animals treated with PKRi (n = 13) or ISRIB (n = 13) was reopened. (c) Visual imprinting was not restored in chickens injected with PKRi (n = 13) and ISRIB (n = 13) or injected with vehicle (n = 12). * indicates p<0.05 from two-ways ANOVA test, Bonferroni *Post hoc* test.

The following source data is available for figure 6:

**Source data 1.** Preference indexes of animals trained in P4 and injected with PKRi, ISRIB or vehicle.

## Imprinting training and preference test

Training sessions and tests were performed in a sound proof chamber (IAC acoustics) at 37°C in the dark, except for the light coming from the monitor. All experiments and drug manipulations were performed blind to treatment. On the training day, each chicken was placed under white light for 30 min. This priming procedure has been extensively used in visual imprinting (*Bolhuis et al., 2000*; *Nakamori et al., 2010*). After priming, chickens were placed in a running wheel (internal diameter = 18 cm) in front of a computer monitor (ACER LCD, 17''). Magnets mounted on the wheel (*Gibbs and Lecanuet, 1981*) allowed the precise measurement of the approaching behavior by a counter (Med Associates, DIG-700G, DIG-726). Each magnet count generated a TTL signal, whose timing was stored in a computer for offline analysis.

Visual stimuli consisted of custom-made animations (Blender, http://www.blender.org/) of either a blue or red rectangular prism coupled to a sound. Both figures had exactly the same volume and followed the same rotation and movement across a virtual room (*Video 1* and *Video 2*). This method made it possible to synthesize arbitrary movement patterns while controlling luminosity, color and shape. Objects changed shape (expansion and contraction) synchronously with sound. Two different sounds were synthesized using Audacity software (Audacity 2.1.0). The frequency range for both sounds was 0–3 KHz. Sound one consisted in frequency steps and sound two was composed of frequency sweeps (see supplementary material). Each sound was played 12 times during a minute, every 3 s. The start of each animation was commanded by software written in Matlab, which was interfaced to Med Associates equipment through a USB DAQ card (National instruments USB-6008).

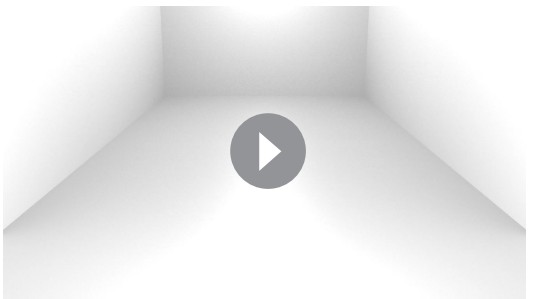 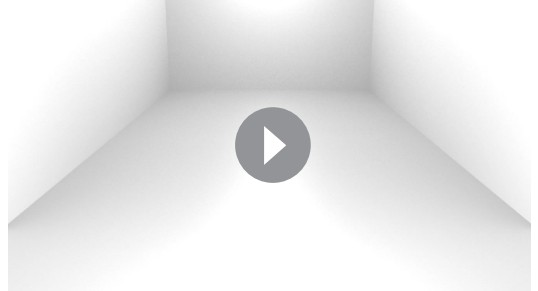

**Video 1.** Stimulus A presented to chickens. This animation was played on a screen during training. For auditory and visual imprinting tests only the auditory or the visual component was presented.

**Video 2.** Stimulus B presented to chickens. This animation was played on a screen during training. For auditory and visual imprinting tests only the auditory or the visual component was presented.

Audiovisual training stimuli were presented in 4 min bouts followed by 1 min of silence and darkness. If the chicken did not move the wheel during the first half hour of exposure, the experiment was interrupted and not included in the sample. Training length varied from 0 to 120 min, depending on the protocol. To investigate long-lasting effects on imprinting we tested chickens the day after training.

Visual and auditory imprinting were tested independently in a sequential test, where the novel and imprinted stimuli are presented in alternation. While other studies have used a simultaneous choice test (*Yamaguchi et al., 2012*), the sequential test allowed us to randomize stimulus presentation, measure baseline locomotion and assess the response to novel and imprinted (*Video 3*) stimuli independently. Each test included 5 presentations of the imprinted stimulus and 5 presentations of the novel stimulus. The duration of each presentation was 1 min. Baseline locomotion was measured during 30 s between trials. Imprinted and novel stimuli were alternated over five consecutive blocks. The first stimulus that started the sequence was picked randomly. Although this method differs from previous reports where fixed sequences were used (*Bolhuis et al., 2000*; *Town and McCabe, 2011*; *McCabe and Horn, 1988*), randomization prevents biases and motivation changes over time emerging from fixed sequences.

Previous studies have used different criteria and indexes to quantify the strength of imprinting. Such quantifications have included differences in time spent in the proximity of the imprinted object (*Yamaguchi et al., 2012*), differences in locomotion toward the imprinted and novel stimulus (*Bolhuis et al., 2000*), differences in locomotion during the presentation of imprinted and novel objects and the absence of a stimulus (*Maekawa et al., 2006*), and number of chickens within a group selecting the imprinted stimulus over several trials (*Wallhäusser and Scheich, 1987*). In this study, we normalized differences between locomotion to novel and imprinted stimuli by the average baseline locomotion in the wheel when no stimulus was presented. Therefore, to assess imprinting, we calculated a preference index (PI), $PI = \sum (Imprinted_{STL} - Novel_{STL}) / Baseline_A$ where *STL* indicates stimulus-triggered locomotion either during the presentation of the imprinted stimulus (*Imprinted_{STL}*) or presentation of the novel stimulus (*Novel_{STL}*), and baseline$_A$ refers to the average baseline locomotion across the experiment. An advantage of this quantification over previous methods is that: (1) it takes into account the fluctuations in basal locomotion before each stimulus presentation, and (2) it weights differences in approaching behavior by average locomotion.

## Assessment of the sensitive period

The sensitive period for filial imprinting has been reported to close within 3–4 days after hatching (*Yamaguchi et al., 2012*). To ensure the training and preference tests captured this sensitivity, the ability of chickens to develop a preference to visual and auditory stimuli within the first 4 days after hatching was measured immediately and 24 hr after training.

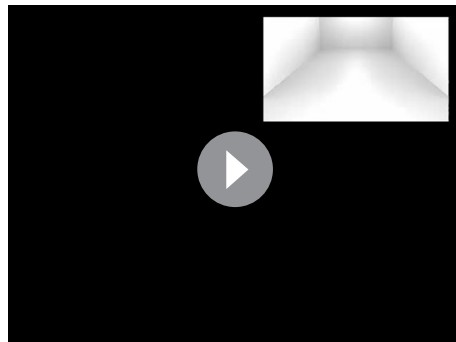

**Video 3.** Chicken imprinted to stimulus B approaching the screen. This approach behavior was quantified during the presentation of imprinted or novel stimuli to compute a preference index.

## In vivo SUnSET

We optimized previously reported in vivo SUnSET protocols in muscle fibers (*Goodman and Hornberger, 2013*; *Goodman et al., 2011*) for monitoring protein synthesis in the chick brain. It has been shown that PMY injected intravenously takes 2–4 hr to be incorporated into the brain (*Flexner et al., 1962*). This contrasts with the fast incorporation (approximately 30 min) into muscle (*Goodman and Hornberger, 2013*; *Goodman et al., 2011*) and other organs (*Flexner et al., 1962*). In pilot experiments, we determined that 3–4 hr after injecting a low dose of PMY (MP Biomedicals, 0.04 mg/g, diluted in distilled $H_2O$, IP) was the optimal time period for detecting the incorporation of PMY in newly synthesized proteins. This information was used to adjust the timing of PMY injection in our behavioral pharmacology experiments.

To simultaneously assess experience-dependent translation across sensory modalities and brain regions, in the same animal, we identified a training schedule that reliably triggered auditory and visual imprinting. Since 2 hr but not 1 hr training (*Figure 3a,b*) triggered both auditory and visual imprinting, we used the former schedule. Four hours after PMY injection chicks were decapitated and brains were rapidly (2–3 min) placed in cold PFA (4%) overnight at 4°C. A vibratome (Leica VT 1000S) was used for making 100 μ sagittal cross sections. After three 10 min washing with PBS, samples were incubated overnight at 4°C in a solution containing antibodies against PMY (EMD Millipore, cat# MABE343, RRID:AB_2566826) and S6 (Cell signaling, cat# 2217, RRID:AB_331355) to identify cell somas. Samples were washed in PBS (three 10 min wash) and placed for 1.5 hr in a solution containing Alexa-488 (Invitrogen, cat# A21202, RRID:AB_2535788) and Alexa-568 (Invitrogen, cat# A10042, RRID:AB_2534017) against the primary antibody host species. After washing again 3 times for 10 min in PBS, samples were covered with Prolong Gold mounting media (Molecular probes, cat# P36935).

A confocal microscope (Zeiss LSM 510 Meta Duo V2) was used to collect images from IMM and MNM (10X, zoom 3). All images were taken blind to the experimental groups. IMM is located 2.5 mm from the dorsal surface of the brain and 0.5–1 mm from the caudal edge of the forebrain, limited below and laterally by the lateral ventricle. MNM is located 0.5–1 mm lateral from the midline, 3 mm from the dorsal surface of the brain and 5 mm from the caudal edge of the forebrain, below the lateral pallial lamina that separates the hyperpallium and mesopallium (*Puelles et al., 2007*). All compared samples were processed the same day, using the same protocol, and images were taken with equal microscope settings. Control animals were housed in the same conditions as trained animals but presented with an empty screen.

Images were analyzed using ImageJ software (NIH, 1.50i). Threshold was adjusted by the S6 signal to select cell somas. PMY signal was detected using the selection created for the S6 channel. To compare across groups all measures were normalized to the average intensity of the control group.

## Protein synthesis inhibition

To investigate the involvement of protein synthesis in long-term memory formation during imprinting we injected cycloheximide (Tocris, CHX, 1 mg/kg, IP), diluted in 0.1% DMSO and saline, immediately after training. Since 1 hr training was enough to generate visual (*Figure 3a*) but not auditory imprinting (*Figure 3b*), we injected CHX immediately after 1 hr and 2 hr training, and tested the effect on imprinting 24 hr later for each sensory modality, independently.

## Manipulation of the eIF2α signaling pathway

We used the specific blocker of eIF2α phosphatases Sal003 (Sigma Aldrich, 0.2 mg/Kg, diluted in 0.1% DMSO and 0.9% Saline, IP) to test whether a reduction in eIF2α phosphorylation is required

for imprinting. We used 2 hr training for this experiment, which reliably triggered strong visual and auditory imprinting, and injected Sal003 before training to ensure translation was inhibited during and immediately after training.

To specifically enhance the formation of imprinted memories by reducing eIF2α–mediated translational control, we conducted two independent manipulations: animals were injected immediately after training with either the specific inhibitor of the eIF2a kinase PKR (PKRi; EMD Millipore, 0.1 mg/Kg, diluted in 0.1% DMSO and 0.9% saline) or ISRIB (Sigma Aldrich, 2.5 mg/Kg, diluted in 50% DMSO and 50% saline, IP), which blocks the translational effect induced by p-eIF2α. To avoid a ceiling effect masking the enhancement of imprinting, we used 1 hr training (weak training) and tested preference 24 hr after training.

## Western blotting

Lysates of IMM and MNM (anatomical boundaries described above) were obtained from brain tissue, punched out from 0.75- to 1mm-thick sagittal brain slices collected from imprinted and control animals. We used antibodies against eIF2α (Cell Signaling Cat #9722, RRID:AB_2230924), p-eIF2α (Ser51)(Cell Signaling Cat #9721, RRID:AB_330951), following standard protocols described before (*Costa-Mattioli et al., 2007*). Control tissue samples were obtained from chickens that ran on the wheel towards a screen displaying only a static image of an empty room, as shown in *Figure 1b* (left panel).

## Reopening of the CP

We tested whether reducing p-eIF2α by PKRi and ISRIB administration could reopen the CP for each sensory modality using 2 hr training on P4. Since injecting PKRi and ISRIB immediately after 1 hr training did not have an effect on visual imprinting, we injected PKRi (*Stern et al., 2013*; *Ingrand et al., 2007*) (0.1 mg/Kg, IP) and ISRIB (2.5 mg/kg) before training to control whether the lack of effect on visual imprinting was due to the time of the injection. Imprinting was assessed 24 hr after training as described above.

## Dendritic spine analysis

Brains were rapidly dissected (in 2–3 min) and placed in paraformaldehyde (4%) for 1 hr, then transferred to the phosphate buffer solution. A vibratome (Leica VT 1000S) was used for making 200 uM slices. Tungsten beads coated with lipophilic dye (DiI) were delivered to each slice using a modified gene gun (*Gan et al., 2000*). The dye was allowed to spread overnight. The next day, each slice was mounted using ProLong Gold mounting media. A confocal microscope (Zeiss LSM 510 Meta Duo V2) was used to collect Z-stacks (63X, zoom 3) from areas of interest containing labeled dendritic branches. Images of secondary branches, within 50–75 μm from the soma, were used for spine analysis.

Dendritic spines were counted blind to experimental groups using Image J software (Version 1.50a). A multicolored lookup table (Fire) was used to reliably visualize individual spines. Two 10 μm segments were marked randomly along each secondary dendritic branch. Spines along each of the two segments were counted by a blind experimenter. The spines 'head width, presence of neck and overall length were used for classifying them in filopodia, stubby, thin, or mushroom, using published criteria (*Bourne and Harris, 2007*; *Sanders et al., 2012*; *Chakravarthy et al., 2006*). Briefly, spines without clear head and neck, and shorter than 1 μm, were categorized as stubby. Spines longer than 1 μm were classified as mushroom or thin, depending on whether a head and neck were observed. Protrusions longer than 2 μm were categorized as filopodia.

To investigate if eIF2α was required for structural plasticity, we injected chickens with Sal003 (i.p., 0.2 mg/kg) and trained them for 2 hr. The day after the training, we labeled dendritic arbors and assessed dendritic spines, as described above.

## Statistical analyses

Statistical analyses were performed using SigmaPlot (Systat Software). Data distribution normality was assessed using the Shapiro-Wilk and F-test to evaluate the differences of variances. When variances were significantly different the Welch's correction was used. Statistics were based on the two-sided Student's t test, or the two-way ANOVA and Bonferroni post-hoc test for multiple

comparisons of normally distributed samples. Otherwise the Mann-Whitney or the Kruskal-Wallis and Dunn's multiple comparisons tests were used. Within-group variation is indicated by standard errors of the mean of each distribution, which are depicted in the graphs as error bars. $p < 0.05$ was considered significant.

## Acknowledgements

We thank Anna Francesconi, Bryen Jordan and Michael Beckert for their critical discussion and comments on the manuscript. We also thank Michael Beckert for helping with illustrations. This study was supported by the Konishi Neuroethology Research Award to GB, by NIH grant number DC007690 and a pilot grant from the Rose F Kennedy Intellectual and Developmental Disabilities Research Center (RFK-IDDRC) to JLP and grants from the National Institutes of Health to MCM (NIMH 096816, NINDS 076708).

## Additional information

### Funding

| Funder | Grant reference number | Author |
|---|---|---|
| International Society for Neuroethology | Konishi Research Award 2016 | Gervasio Batista |
| National Institutes of Health | NIMH 096816 | Mauro Costa-Mattioli |
| National Institutes of Health | NINDS 076708 | Mauro Costa-Mattioli |
| National Institutes of Health | DC007690 | Jose L Pena |
| Rose F. Kennedy Intellectual and Developmental Disabilities Research Center | U54 HD090260 | Jose L Pena |

The funders had no role in study design, data collection and interpretation, or the decision to submit the work for publication.

### Author contributions

GB, Conceptualization, Data curation, Formal analysis, Funding acquisition, Investigation, Visualization, Methodology, Writing—original draft, Writing—review and editing, Conceived and designed the study, Performed all behavioral experiments, extraction of tissue samples and diolistic labeling, Designed the p-eIF2α-mediated translation loss- and gain- of function experiments, Helped with the interpretation of the behavioral and western blotting results, Wrote the manuscript with input from all other authors, Conception and design, Acquisition of data, Analysis and interpretation of data, Drafting or revising the article; JLJ, Investigation, Performed all the western blots; ED, Investigation, Counted, measured and classified dendritic spines; MC-M, Conceptualization, Formal analysis, Supervision, Methodology, Writing—review and editing, Designed the p-eIF2α-mediated translation loss- and gain- of function experiments, Helped with the interpretation of the behavioral and western blotting results, Wrote the manuscript with input from all other authors; JLP, Conceptualization, Formal analysis, Supervision, Funding acquisition, Investigation, Methodology, Writing—original draft, Project administration, Writing—review and editing, Conceived and designed the study, Directed and supervised the experiments, Wrote the manuscript with input from all other authors

### Author ORCIDs

Gervasio Batista, http://orcid.org/0000-0003-0885-1224
Mauro Costa-Mattioli, http://orcid.org/0000-0002-9809-4732

### Ethics

Animal experimentation: Experiments and euthanasia method were approved by the institutional animal care committee (IACUC) at Albert Einstein College of Medicine (protocol 20140910).

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
