## [Decision Letter]

[Editors’ note: this article was originally rejected after discussions between the reviewers, but the authors were invited to resubmit after an appeal against the decision.]

Thank you for submitting your work entitled "Translational control of auditory imprinting and structural plasticity by eIF2α" for consideration by *eLife*. Your article has been reviewed by three peer reviewers, and the evaluation has been overseen by a Reviewing Editor and Eve Marder as the Senior Editor. The reviewers have opted to remain anonymous.

Our decision has been reached after consultation between the reviewers. Based on these discussions and the individual reviews below, we regret to inform you that your work will not be considered further for publication in *eLife*.

The question of what regulates critical periods and imprinting has broad appeal, and your efforts to show a modality specific translational control mechanism underlying the critical period for imprinting were of great interest to the reviewers. As currently presented, however, the reviewers did not find your conclusions that the translational regulator eIF2α and its phosphorylation state regulate the formation of auditory memories and spine plasticity convincing. For the protein synthesis aspect, better demonstration of actual alterations in synthesis of given proteins would strengthen the study. In addition, more clearly stating the time points of injection relative to the timing of training, confirmation of the brain areas examined for sampling of spine changes, and clarification of whether the p-eIF2α signal derived from neurons or from neurons and glia by immunohistochemistry of the tissue taken for immunoblotting, would enhance your presentation.

Your attempts to distinguish effects on auditory but not visual imprinting are laudable, but are confounded by difficulties in the field in separating stimuli for auditory versus visual learning; your design should be better argued compared to previous efforts. Finally, the novel method you use for the preference index for analyzing imprinting could be better explained and compared to previously used tests, and the appropriateness of the statistical tests you used considered.

Reviewer #1:

The work by Batista et al. is very intriguing. Understanding what regulates critical periods and imprinting are indeed important questions that need to be addressed and will be of great interest to a broad community. While the title suggests that eIF2α controls translation-dependent auditory imprinting and structural plasticity, the authors do not provide any direct evidence that new proteins are synthesized. At best, the authors can claim that eIF2α is involved in auditory imprinting. Thus, the authors need to provide convincing evidence that translation, or increased protein synthesis, is required for auditory imprinting and structural plasticity. In addition, while the literature suggests eIF2α is involved in memory formation, so are other protein synthesis pathways such as mTOR, as mentioned in the Discussion. The authors should include Western blot analysis for other protein synthesis pathways, as a control.

Major comments:

Figure 2

The use of cycloheximide does not indicate that protein synthesis is required. To show convincingly that protein synthesis is involved, the authors must show that (1) total protein synthesis (or some proteins) is elevated at 2-hour training in auditory imprinting and at 1-hour training in visual imprinting, and (2) training-induced increase in protein synthesis is blocked by cycloheximide. This can be done by metabolic labeling or demonstration that levels of select proteins are increased by Western blot analysis.

Figure 3

Total eIF2α seems to be reduced in imprinted samples. The authors should measure other proteins whose levels are not affected by imprinting and normalize total eIF2α to demonstrate that imprinting affects eIF2α phosphorylation and not total levels of eIF2α.

The authors should also show that the manipulation of eIF2α phosphorylation directly affects protein synthesis. They may demonstrate this by measuring total protein (or select proteins) levels of MNM.

Figure 5

The authors use PKRi and ISRIB to investigate the critical period. It's rather dubious that Sal003 was not used to investigate the critical period. Does Sal003 fail to open the critical period? The authors should use the same manipulation/drugs for Figure 4 and Figure 5.

Reviewer #2:

This manuscript addresses an important issue in the molecular mechanism of imprinted memories during a critical period. The authors argue that the translational control by phosphorylation of the eukaryotic translation-initiation factor 2α subunit (eIF2α) is required for auditory imprinting, and mediates changes in dendritic spines in the MNM region. They present data that the increasing phosphorylated eIF2α prevents the formation of auditory memories and spine plasticity. In contrast, the inhibition of an eIF2α kinase enhances auditory memories. Furthermore, they provide evidence that the blocking p-eIF2α mediated translational control enhances auditory memories and reopens the critical period. They propose a modality-specific translational control mechanism underlying the critical period for imprinting.

The paper contains sufficient interest and originality to merit publication, however, the impact is weakened by a technical limitation of the methods used to quantify the preference score and to analyze the statistics. While the findings presented in this study should be of interest to the readers in the field of animal behavior, it is my opinion that a rather substantial revision, based on the technical comments given below, is needed to make this manuscript suitable for publication.

Major points

My main concern is that the preference index of the authors (<imprinting-control>/baseline) differs from that of the conventional method of Horn et al. (imprinting/<imprinting + control>)(P14 L231). Did they use the background on the screen as the "baseline" image? Compared to the method of Horn et al., this method seems to be affected by the individual difference of animals. I am wondering why the authors used "baseline". The authors should clarify the reason why each score is divided by the baseline. The authors may hypothesize that more active chicks in locomotion should show bigger number of preference score than inactive chicks. But the less active chicks show the highest preference in some cases. If the data are analyzed by the method of Horn et al., it will be easier for most of the readers in the field of imprinting research to understand the significance of this paper.

As for the statistical analysis, the authors use either unpaired t-test or Mann-Whitney U test to examine the difference between two groups. The statistical tests were not performed appropriately in some cases. For example, Mann-Whitney U test should be used in the data of Figure 3 (Auditory imprinting) because the variances between two groups in Figure 3 are significantly different (F-test, p=0.0253), but the authors use unpaired t-test. On the other hand, unpaired t-test should be used for data of Figure 3, because the variances between two groups were not different (F-test, ns). The authors should examine the difference of variances by F-test in all data to choose the appropriate statistical test.

There is controversy whether there is genuine auditory imprinting (Bolhuis, J.J. et al., Behaviour Vol. 122, (1992), pp.195-230). It is generally concluded that auditory stimuli play an important role in the formation of filial preferences, but that auditory imprinting is not as prominent when compared to visual imprinting. Although a lot of studies have demonstrated that the addition of an auditory stimulus improves following of a visual stimulus, few studies using only auditory exposure have demonstrated significant auditory learning. As is the case in this paper, the majority of studies of early auditory learning have used a compound of a visual and an auditory stimulus during training. The authors should discuss the physiological significance of auditory imprinting in relation to visual imprinting in the Introduction section.

Figure 2

I suggest the chemicals like cycloheximide should be used by direct injection into the target brain area, not by intraperitoneal injection because cycloheximide is a potent inhibitor of translation and therefore will be toxic to any kind of cells in the whole body.

Figure 3

In western blotting, the time course experiment at different time points (before training, during training, 2-3 h after training, et al.) is necessary to know whether the decrease of p-eIF2α is temporary or not. Is there any difference in the amount of p-eIF2α in the MNM region between 1 day old and 4 day old control chick?

Did the authors examine the decrease of p-eIF2α was learning dependent or locomotion dependent? The authors may know the answer by doing experiments under the condition of forced running.

As the authors punched out the region of MNM or IMM out of brain slice section, they should mention the exact location in the brain for the extraction in detail in the methods section. This definition of the brain area is very important because either the MNM or IMM region does not have the clear border from surrounding areas. Was the decrease of p-eIF2α detected specifically in the MNM region of the brain?

Figure 4

I understand the imaging study takes a lot of effort to detect the change of the properties of dendritic spines. I assume the neurons are randomly labeled in the MNM. Therefore, more than 1000 spines at least per each experimental setting should be examined to detect significant morphological changes beyond individual deviation. At the same time, the exact location of brain area they examined and the morphological definition of dendritic spine for classification in chick should be mentioned in detail in the method section

Reviewer #3:

This manuscript from Batista et al. is beautifully written and addresses the role of translational regulation during critical period plasticity in the chick. The study includes data indicating that auditory imprinting involves translation initiation regulation by eIF2α, and in particular that eIF2α is phosphorylated and inhibited to allow for plasticity in the auditory cortex during the critical period. This indicates that mRNAs containing upstream open reading frames (whose translation is upregulated when eIF2α is phosphorylated and inhibited) are of particular importance to auditory plasticity during the critical period. In contrast, the authors show that phosphorylation of eIF2α, and hence eIF2α translational regulation, is not involved in visual imprinting. The findings are intriguing and I believe important. However, I also believe that there are some significant flaws in the experimental design and interpretation that weaken the study. My principal concerns have to do with the question of how well the translational inhibition worked, and the question of the kinetics of translational regulation in the MNM (for auditory imprinting) and IMM (for visual imprinting). What were the kinetics of inhibition of protein synthesis following IP injection? Was it equivalent in MNM and IMM? It's not clear from the manuscript whether or not the IP injection was done at the end of the 1 hr and the 2 hr training. The authors write: "both auditory and visual imprinting rely on protein synthesis but following different temporal dynamics", and yet the conclusions from the translational inhibition experiments are all based on manipulations that do not appear to consider the differential temporal dynamics of imprinting (because they simply target a single time point, optimized for the effects on auditory imprinting). I also think it's really important to show that there is an effect on translation, and in particular on eIF2α-mediated translation. This is provided for Sal003 in Figure 3—figure supplement 1, but only for "brain samples," not for IMM or MNM. Without this, the results are all based on pharmacological interventions, which always have some caveats unless the expected effects are directly measured. For example it looks like there is significant variability in the ratio of peIF2α to eIF2α in Figure 3—figure supplement 1, which raises the question of whether this correlates in any way with plasticity during imprinting.

[Editors’ note: what now follows is the decision letter after the authors submitted for further consideration.]

Thank you for resubmitting your work entitled "Translational control of auditory imprinting and structural plasticity by eIF2α" for further consideration at *eLife*. Your revised article has been favorably evaluated by Eve Marder as the Senior editor, a Reviewing editor, and two reviewers.

The manuscript has been improved but there are some remaining issues that need to be addressed before acceptance, as outlined below:

The reviewers have several queries, and ask for further clarification on the following:

Reviewer 1 asks for you to explain why protein synthesis dependent visual plasticity at 1 hour of training was ignored and the focus placed on 2 hours; an explanation would help the reader move to the logic of the next set of experiments.

In addition, this reviewer wanted an explanation of the relevance of structural spine changes over density in the context of behavior.

Reviewer 2 wondered why the first trials of the novel stimulus strongly suppressed locomotion; this does not seem to occur in the reviewer's hands in their experimental design. This reviewer is concerned about whether the training for imprinting by the authors is sufficient to cause strong preference compared with controls.

Here are the comments in greater detail:

Reviewer #1:

The manuscript "Translational control of auditory imprinting and structural plasticity by eIF2α" by Batista et al. is greatly improved. This is an important manuscript describing how protein synthesis mediates structural plasticity and behavior. The finding that restoring translational control of eIF2α opens the critical period is timely and important. The authors have addressed most of my concerns.

Still missing from the manuscript is an explanation as to why the authors ignored the protein synthesis dependent visual plasticity at 1 hour of training. A sentence or two justifying the shift toward the 2 hour training period and the focus on auditory imprinting over visual imprinting would provide the reader with the logic behind the next set of experiments. Also, further discussion on the relevance of the structural spine changes over density in the context of behavior would strengthen the manuscript (see Bourne and Harris, Hippocampus, 2011).

Reviewer #2:

The revised manuscript is a great improvement on the original and now suitable for publication.

I will just add some comments about the authors' preference index in imprinting. I am a little bit surprised to hear that the first trials of the novel stimulus generated a stronger suppression of locomotion because it never happens when we use the simultaneous choice test as a method to measure the preference for visual imprinting. It seems to me that the novel stimulus causes fear for chicks in the initial trials. Usually, newly-hatched chicks show the preference to novel conspicuous moving objects. Rather in some cases, I even choose the object which attracts the intrinsic preference of chicks as a control in the test. Under the condition, the imprinted chicks still show the strong preference against the controls in the simultaneous choice test. I am worried whether or not the training for imprinting by the authors is sufficient enough to cause the strong preference against the controls. Also, there is a possibility that the increasing number of trials in the test can bring the similar effect as the imprinting training. The authors may improve the methods of the training and the test in future to strengthen their interesting findings by the molecular approach.

---

## [Author Response]

[Editors’ note: the author responses to the first round of peer review follow.]

*The question of what regulates critical periods and imprinting has broad appeal, and your efforts to show a modality specific translational control mechanism underlying the critical period for imprinting were of great interest to the reviewers. As currently presented, however, the reviewers did not find your conclusions that the translational regulator eIF2α and its phosphorylation state regulate the formation of auditory memories and spine plasticity convincing. For the protein synthesis aspect, better demonstration of actual alterations in synthesis of given proteins would strengthen the study.*

As requested by the editor, we now measured protein synthesis in vivoin response to drug treatments and training using the SUrface SEnsing of Translation (SUnSET) technique. We now show that imprinting triggers translation in imprinting-relevant areas of the forebrain (new Figure 2) and validate all pharmacological manipulations on the eIF2α pathway as means of regulating translation (new Figure 4—figure supplement 2). These new results strengthen our conclusions on the translational control of auditory imprinting by eIF2α.

*In addition, more clearly stating the time points of injection relative to the timing of training, confirmation of the brain areas examined for sampling of spine changes, and clarification of whether the p-eIF2α signal derived from neurons or from neurons and glia by immunohistochemistry of the tissue taken for immunoblotting, would enhance your presentation.*

To clarify the time points of injection relative to the timing of training we added a schematic in each figure indicating when injection was performed relative to training. Also, in the methods section we provide the rationale for the timing of the injection for each experiment.

As pointed out by one of the reviewers, the boundaries of MNM and IMM are diffuse, thus now we report the coordinates used in our study to sample spine changes within those regions. In brief, since there are no- specific markers for each area, we determined the location of MNM and IMM based on anatomical landmarks. Also, while these structures are reported to be large (~1 mm^3^), we constrained our search for cells within a 0.5 mm radius around the center.

eIF2α is ubiquitous across cell types, therefore we find interesting the question whether experience-dependent changes in eIF2α phosphorylation emerge from glia or neurons. Indeed, even in the memory field the cell-type contribution of eIF2α phosphorylation remains to be determined. in vitrostudies have shown that changes in glial eIF2α after glutamate application are transient, lasting around 5-10 minutes. It has been speculated that this process helps glial cells to cope with the metabolic stress of removing glutamate from the synaptic cleft. We included this information at the end of the third paragraph of the Discussion section and raised this interesting and important question for future work.

*Your attempts to distinguish effects on auditory but not visual imprinting are laudable, but are confounded by difficulties in the field in separating stimuli for auditory versus visual learning; your design should be better argued compared to previous efforts.*

It has been shown audiovisual stimulation enhances both visual and auditory imprinting tested separately (Bolhuis et al., 1992). Moreover, investigations on visual imprinting have commonly used audiovisual stimulation during training. However, most studies have focused on the mechanisms underlying visual imprinting because studying auditory imprinting carries additional experimental challenges, such as effect of previous social interaction and the difficulty in eliciting robust auditory memory. To clarify how we overcame these difficulties we revised the second paragraph of the Discussion section. In brief, our behavioral paradigm aimed to: 1) restrict social interaction and raise chicks in darkness, 2) generate a balanced data set where chicks were trained with two different synthetic sounds, and 3) increase the training length compared to previous studies to generate a robust long-lasting auditory memory.

*Finally, the novel method you use for the preference index for analyzing imprinting could be better explained and compared to previously used tests, and the appropriateness of the statistical tests you used considered.*

Imprinting strength has been assessed differently across studies. We now mention the different approaches used in the past before introducing ours. As pointed out by one of the reviewers, the classic preference score (PS) was the ratio between locomotion during imprinted trials and total locomotion, PS= locomotion during imprinted trials / total locomotion. This preference index was used previously to assess imprinting strength in a 4-trial testing paradigm with stimuli presented in ‘fixed’ sequence (either imprinted- novel-novel-imprinted or novel-imprinted-imprinted-novel). Upon preliminary analysis, we noticed that considering only the first 4 trials for computing the classic preference score overestimated the strength of the imprinting (Figure 7). This was mainly because the first trials of the novel stimulus generated a stronger suppression of locomotion, compared to subsequent presentations of the same stimulus. For this reason, we decided to test imprinting strength in 10 pseudo-random trials. In addition, total locomotion varied largely from first to last trials mainly because locomotion during the novel trials increased over time (Figure 7). Because baseline locomotion was the most stable parameter, on average, we decided to use it to normalize the difference in locomotion between imprinted and novel trials. We believe our computation of the preference score is advantageous for randomizing the stimulus presentation (thus ruling out biases due to sequence), normalizing by a more stable variable (baseline) and increasing the number of trials to 10 to assess not only novelty detection but also the maintenance of the preference over time.

Author response image 1.) Classic preference score (PS) for auditory and visual imprinting computed as PS=locomotion during imprinted trials/ total locomotion, in chickens trained for 2 hours (n=13), b) locomotion before and during the presentation of the imprinted stimulus (n=26, in each trial) and c) locomotion before and during the presentation of the novel stimulus (n=26, in each trial)**DOI:**
http://dx.doi.org/10.7554/eLife.17197.020

We appreciate the comments on the statistical analysis. We revised the tests used, considering the differences in variance between groups, and applied the appropriate tests for each comparison. The statistical significance persisted after the revised analysis.

In summary, we now show that: a) training triggers an increase in translation in vivothat can be blocked with cycloheximide, b) translation is decreased in animals treated with Sal003, which increases eIF2α phosphorylation and blocks auditory imprinting, and c) translation is increased in animals treated with ISRIB, an enhancer of eIF2α- dependent translation that reopens the critical period for auditory imprinting. In addition, we clarified the timing of the injection, included a discussion of the putative role of p-eIF2α translational control in glial and justified our experimental design with reference to previous studies and a detailed analysis of the chickens’ behavior.

*Reviewer #1:*

*The work by Batista et al. is very intriguing. Understanding what regulates critical periods and imprinting are indeed important questions that need to be addressed and will be of great interest to a broad community. While the title suggests that eIF2α controls translation-dependent auditory imprinting and structural plasticity, the authors do not provide any direct evidence that new proteins are synthesized. At best, the authors can claim that eIF2α is involved in auditory imprinting. Thus, the authors need to provide convincing evidence that translation, or increased protein synthesis, is required for auditory imprinting and structural plasticity. In addition, while the literature suggests eIF2α is involved in memory formation, so are other protein synthesis pathways such as mTOR, as mentioned in the Discussion. The authors should include Western blot analysis for other protein synthesis pathways, as a control.*

We thank the reviewer for this comment. As requested, we now measure translation in vivoin response to training and tested the effect of different drugs. As shown in the new Figure 2, training increased translation in imprinting-relevant areas. In addition, we demonstrate that this increase in translation can be blocked with cycloheximide (Figure 3). Also, we validated our pharmacological manipulations and demonstrated that Sal003 reduces and ISRIB enhances protein synthesis in vivo(new Figure 4—figure supplement 2).

We acknowledge that other protein synthesis pathways such as mTORC1 are also involved in memory formation. However, very little is known about the translational control pathways involved in imprinting. In this paper we provided gain- and loss- of function evidence that the translational program mediated by eIF2α is crucial for imprinting. The role of other translational pathways in imprinting is currently under investigation by our group.

*Major comments:*

Figure 2

*The use of cycloheximide does not indicate that protein synthesis is required. To show convincingly that protein synthesis is involved, the authors must show that (1) total protein synthesis (or some proteins) is elevated at 2-hour training in auditory imprinting and at 1-hour training in visual imprinting, and (2) training-induced increase in protein synthesis is blocked by cycloheximide. This can be done by metabolic labeling or demonstration that levels of select proteins are increased by Western blot analysis.*

Using SUnSET we now show that training induces an increase in protein synthesis (see new Figure 2). Moreover, we also show that this experience-dependent increased in translation is blocked by CHX (Figure 3)

Figure 3

*Total eIF2α seems to be reduced in imprinted samples. The authors should measure other proteins whose levels are not affected by imprinting and normalize total eIF2α to demonstrate that imprinting affects eIF2α phosphorylation and not total levels of eIF2α.*

We performed western blots against total eIF2α and normalized to ß-tubulin to address this concern. As shown in Figure 8, there are no significant differences in the total levels of eIF2α between untrained and trained animals.

Author response image 2.) total eIF2α levels normalized to ß-tubulin levels show no difference across untrained (n=7) and trained (n=6) animals, b) representative images of western blots performed for quantification.**DOI:**
http://dx.doi.org/10.7554/eLife.17197.021

*The authors should also show that the manipulation of eIF2α phosphorylation directly affects protein synthesis. They may demonstrate this by measuring total protein (or select proteins) levels of MNM.*

As requested by the reviewer, we now show that we can bidirectionally manipulate translation rates using either ISRIB or Sal003 (Figure 4—figure supplement 2).

Figure 5

*The authors use PKRi and ISRIB to investigate the critical period. It's rather dubious that Sal003 was not used to investigate the critical period. Does Sal003 fail to open the critical period? The authors should use the same manipulation/drugs for Figure 4 and Figure 5.*

We apologize for the lack of clarity regarding the action of Sal003. We have included a schematic showing that Sal003 inhibits the PP1-GADD34, PP1-Crep complexes. GADD34 and Crep are PP1 cofactors that render the phosphatase specific to eIF2α. Thus, we show that Sal003-treatment increases eIF2α phosphorylation (new Figure 4—figure supplement 1), reduces protein synthesis (new Figure 4—figure supplement 2) and blocks auditory imprinting (new Figure 4) during the critical period. Therefore, Sal003 was not expected to re-open the critical period.

*Reviewer #2:*

*This manuscript addresses an important issue in the molecular mechanism of imprinted memories during a critical period. The authors argue that the translational control by phosphorylation of the eukaryotic translation-initiation factor 2 α subunit (eIF2 α) is required for auditory imprinting, and mediates changes in dendritic spines in the MNM region. They present data that the increasing phosphorylated eIF2 α prevents the formation of auditory memories and spine plasticity. In contrast, the inhibition of an eIF2 α kinase enhances auditory memories. Furthermore, they provide evidence that the blocking p-eIF2 α mediated translational control enhances auditory memories and reopens the critical period. They propose a modality-specific translational control mechanism underlying the critical period for imprinting.*

*The paper contains sufficient interest and originality to merit publication, however, the impact is weakened by a technical limitation of the methods used to quantify the preference score and to analyze the statistics. While the findings presented in this study should be of interest to the readers in the field of animal behavior, it is my opinion that a rather substantial revision, based on the technical comments given below, is needed to make this manuscript suitable for publication.*

We thank reviewer #2 for the comments. We agree that our computation of the preference index should be more detailed and justified. As stated above, imprinting preference scores are influenced both by novelty and persistence of a preference for the imprinted stimulus. Thus, we designed our index taking both components into account, to avoid biasing the scores by disproportionate differences in the first trials due to novelty responses. The new version of the manuscript elaborates on this issue and compares our index with the ones in previous studies.

*Major points*

*My main concern is that the preference index of the authors (<imprinting-control>/baseline) differs from that of the conventional method of Horn et al. (imprinting/<imprinting + control>)(P14 L231). Did they use the background on the screen as the "baseline" image? Compared to the method of Horn et al., this method seems to be affected by the individual difference of animals. I am wondering why the authors used "baseline". The authors should clarify the reason why each score is divided by the baseline. The authors may hypothesize that more active chicks in locomotion should show bigger number of preference score than inactive chicks. But the less active chicks show the highest preference in some cases. If the data are analyzed by the method of Horn et al., it will be easier for most of the readers in the field of imprinting research to understand the significance of this paper.*

We clarified and justified our metric for assessing imprinting strength. This index was generated to reflect imprinting across a larger number of trials and obtain a more accurate measure of memory strength. In contrast with previous studies, we did not use fixed presentation sequences and opted to pseudo randomize presentation order. We would like to argue that this paradigm better controls for the possible effect of stimulus presentation order. In addition, a fine trial-by-trial analysis of the locomotor response to novel and imprinted stimuli showed that the preference score captures the influence of both novelty detection and the persistence of increased locomotion towards the imprinted stimulus. Therefore, while classic preference scores are computed over 4 trials, we decided to measure preference across 10 trials. Based on the same analysis, we noticed that baseline (locomotion when only a background image is presented), was the most stable parameter during the experiment. Thus we decided to normalize differences to baseline.

*As for the statistical analysis, the authors use either unpaired t-test or Mann-Whitney U test to examine the difference between two groups. The statistical tests were not performed appropriately in some cases. For example, Mann-Whitney U test should be used in the data of Figure 3 (Auditory imprinting) because the variances between two groups in Figure 3 are significantly different (F-test, p=0.0253), but the authors use unpaired t-test. On the other hand, unpaired t-test should be used for data of Figure 3, because the variances between two groups were not different (F-test, ns). The authors should examine the difference of variances by F-test in all data to choose the appropriate statistical test.*

We specially thank the reviewer for this comment. Now, in addition to whether or not the data fit a normal distribution, we also selected the statistical test that was the most appropriate based on differences in variances. When an unpaired t-test was used and statistically significant differences in variances were detected, Welch’s correction was applied as in new Figure 4 (Figure 3 in previous version).

*There is controversy whether there is genuine auditory imprinting (Bolhuis, J.J. et al., Behaviour Vol. 122, (1992), pp.195-230). It is generally concluded that auditory stimuli play an important role in the formation of filial preferences, but that auditory imprinting is not as prominent when compared to visual imprinting. Although a lot of studies have demonstrated that the addition of an auditory stimulus improves following of a visual stimulus, few studies using only auditory exposure have demonstrated significant auditory learning. As is the case in this paper, the majority of studies of early auditory learning have used a compound of a visual and an auditory stimulus during training. The authors should discuss the physiological significance of auditory imprinting in relation to visual imprinting in the Introduction section.*

The definition of imprinting has evolved to not only comprise the acquired social preference of precocial birds but also a variety of early learning processes occurring during a critical period. Based on this, we defined auditory imprinting as a memory formation process that occurs exclusively during the critical period. However, to address the critiques raised by Bolhuis, J.J. et al., Behaviour Vol. 122, we followed a stringent experimental design that is now described in the second paragraph of the Discussion section.

Figure 2

*I suggest the chemicals like cycloheximide should be used by direct injection into the target brain area, not by intraperitoneal injection because cycloheximide is a potent inhibitor of translation and therefore will be toxic to any kind of cells in the whole body.*

We are aware of the limitations of systemic injections but we think it is adequate for most of our experiments since we are looking into differences across brain regions and sensory modalities. We reasoned that an approach where drugs access both regions at a similar rate from a unique source facilitates the interpretation of the results. In addition, local injections in MNM and in IMM are relatively close. Thus, spatial control of local injections would be unreliable.

A toxic effect of cycloheximide is unlikely. For example, in new Figure 3, cycloheximide had no effect on auditory imprinting but specifically inhibited visual imprinting, demonstrating that even when cycloheximide was administered, chicks could perform correctly the visual task and arguing against a toxicity confound.

Figure 3

*In western blotting, the time course experiment at different time points (before training, during training, 2-3 h after training, et al.) is necessary to know whether the decrease of p-eIF2α is temporary or not. Is there any difference in the amount of p-eIF2α in the MNM region between 1 day old and 4 day old control chick?*

To address the temporal dynamics of eIF2α phosphorylation we performed new western blots from 1- day old and 4-days old brain samples and found no differences across developmental stages (Figure 9). This result was very insightful since it indicates that the closure of the critical period is not achieved through an increase in eIF2α phosphorylation. It is likely that a mechanism upstream eIF2α is responsible for closing the critical period. Future investigation will address this interesting question.

Author response image 3.Quantification of p-eIF2α/eIF2α ration between P1 and P4 in MNM (blue, n_P1_=7, n_P4_=7) and IMM (green, n_P1_=7, n_P4_=7).No significance was detected in both areas across developmental time points.**DOI:**
http://dx.doi.org/10.7554/eLife.17197.022

*Did the authors examine the decrease of p-eIF2 α was learning dependent or locomotion dependent? The authors may know the answer by doing experiments under the condition of forced running.*

We apologize for not making this point clear. All the controls were animals that ran towards a screen containing an empty background image without sound.

*As the authors punched out the region of MNM or IMM out of brain slice section, they should mention the exact location in the brain for the extraction in detail in the methods section. This definition of the brain area is very important because either the MNM or IMM region does not have the clear border from surrounding areas. Was the decrease of p-eIF2α detected specifically in the MNM region of the brain?*

To further confirm the specificity of eIF2α dephosphorylation we performed western blots of p-eIF2α and eIF2α in the caudolateral nidopalium (NCM), an area involved in auditory learning in songbirds (Bolhuis et al., Eur J Neurosc., 2001). We did not find significant changes in eIF2α phosphorylation after training in this area (Figure 10).

Author response image 4.Quantification of p-eIF2α/eIF2α ratio in NCM comparing untrained (n=7) and trained (n=7) chickens.No significant difference was detected between both areas across treatments.**DOI:**
http://dx.doi.org/10.7554/eLife.17197.023

Figure 4

*I understand the imaging study takes a lot of effort to detect the change of the properties of dendritic spines. I assume the neurons are randomly labeled in the MNM. Therefore, more than 1000 spines at least per each experimental setting should be examined to detect significant morphological changes beyond individual deviation. At the same time, the exact location of brain area they examined and the morphological definition of dendritic spine for classification in chick should be mentioned in detail in the method section*

The analysis method of dendritic spines is an important issue that we considered before performing the experiments. One advantage of the ‘diolistic’ technique is that it granted stochastic labeling of neurons both in MNM and IMM. Thus the results were not affected by labeling biases.

We agree that a large number of spines and dendrites should be analyzed to detect biologically meaningful differences between experimental groups. To assess this we carried out a power analysis to determine the minimum sample size per group (n_dendrites_=42, for statistical power= 0.85) and aimed to a total number of spines per group (average for our study= 673.33) within the range of previous studies in birds (~450-1500 spines, Roberts et al., 2010). In all analyzed groups our sample size is above 42 dendrites.

As mentioned above, we now included a more detailed description of the coordinates of the areas. We also include a more detailed description of the criteria in the methods section and a schematic of the categories in the new Figure 5.

*Reviewer #3:*

*This manuscript from Batista et al. is beautifully written and addresses the role of translational regulation during critical period plasticity in the chick. The study includes data indicating that auditory imprinting involves translation initiation regulation by eIF2α, and in particular that eIF2α is phosphorylated and inhibited to allow for plasticity in the auditory cortex during the critical period. This indicates that mRNAs containing upstream open reading frames (whose translation is upregulated when eIF2α is phosphorylated and inhibited) are of particular importance to auditory plasticity during the critical period. In contrast, the authors show that phosphorylation of eIF2α, and hence eIF2α translational regulation, is not involved in visual imprinting. The findings are intriguing and I believe important. However, I also believe that there are some significant flaws in the experimental design and interpretation that weaken the study. My principal concerns have to do with the question of how well the translational inhibition worked, and the question of the kinetics of translational regulation in the MNM (for auditory imprinting) and IMM (for visual imprinting). What were the kinetics of inhibition of protein synthesis following IP injection? Was it equivalent in MNM and IMM? It's not clear from the manuscript whether or not the IP injection was done at the end of the 1 hr and the 2 hr training. The authors write: "both auditory and visual imprinting rely on protein synthesis but following different temporal dynamics", and yet the conclusions from the translational inhibition experiments are all based on manipulations that do not appear to consider the differential temporal dynamics of imprinting (because they simply target a single time point, optimized for the effects on auditory imprinting). I also think it's really important to show that there is an effect on translation, and in particular on eIF2α-mediated translation.*

We agree with the reviewer that ensuring that drugs have a similar temporal dynamic to alter protein synthesis is crucial for our interpretations. Using SUnSET, we now show that we can alter translation bidirectionally, in both IMM and MNM, with pharmacology (new Figure 4—figure supplement 2).

*This is provided for Sal003 in Figure 3—figure supplement 1, but only for "brain samples," not for IMM or MNM. Without this, the results are all based on pharmacological interventions, which always have some caveats unless the expected effects are directly measured. For example it looks like there is significant variability in the ratio of peIF2α to eIF2α in Figure 3—figure supplement 1, which raises the question of whether this correlates in any way with plasticity during imprinting.*

There is some intrinsic variability in p-eIF2α levels that cannot be correlated with performance, so far, because to carry out such experiments, quantification of p-eIF2α needs to be done before testing the animals. in vivomeasures of p-eIF2α, which are required to overcome this difficulty, are not yet available.

[Editors’ note: the author responses to the re-review follow.]

*[…] The reviewers have several queries, and ask for further clarification on the following:*

*Reviewer 1 asks for you to explain why protein synthesis dependent visual plasticity at 1 hour of training was ignored and the focus placed on 2 hours; an explanation would help the reader move to the logic of the next set of experiments.*

We decided to use 2-hour training to evaluate experience-dependent changes in translation because it allowed us to assess protein synthesis across brain regions in the same animal. Using 1-hour training did not induce reliable auditory imprinting. The interesting difference in time course is conveyed to readers in the Results section ‘Protein-synthesis dependency of auditory and visual imprinting (Figure 3). We now further clarify the motivation for the 2-hour training in the methods as follows:

“To simultaneously assess experience-dependent translation across sensory modalities and brain regions, in the same animal, we identified a training schedule that reliably triggered auditory and visual imprinting. Since 2-hour but not 1-hour training (Figure 3) triggered both auditory and visual imprinting, we used the former schedule.”

*In addition, this reviewer wanted an explanation of the relevance of structural spine changes over density in the context of behavior.*

We thank Reviewer 1 for suggesting references, which we now cite. To elaborate on the possible relevance of the observed structural plasticity on behavior and memory formation, we added the following paragraph to the Discussion:

“Different forms of structural plasticity have been linked to memory, including spine turnover and morphological changes of preexisting spines (32). In this case, the structural plasticity found in IMM and MNM could be consistent with potentiation and enlargement of specific dendritic spines, favoring the detection of imprinted stimuli. While we did not observe changes in spine density, the increase in mushroom spines and decrease in thin spines may suggest coordinated structural plasticity as previously reported in hippocampal slices (42)”

*Reviewer 2 wondered why the first trials of the novel stimulus strongly suppressed locomotion; this does not seem to occur in the reviewer's hands in their experimental design. This reviewer is concerned about whether the training for imprinting by the authors is sufficient to cause strong preference compared with controls.*

We thank Reviewer 2 for sharing personal experience.

We chose the sequential scheme in a running wheel because it allowed us to: 1- randomize the stimulus order, 2- measure baseline locomotion, 3- assess independently the response to novelty and to the imprinted stimulus. We have not been the first using this training schedule and reporting suppression by novel stimuli. Initial suppression of locomotion by the novel stimulus using a similar setup to ours has been reported (Figure 1; Maekawa et al., 2006). In addition, Maekawa et al. (2006) showed that aversion to novelty emerged after training. Thus behavioral suppression by novel stimuli is not inconsistent with innate attraction to conspicuous moving objects. To contextualize our methodology we now highlight in the methods our use of a sequential test and acknowledge the simultaneous choice test, used in other studies:

“Visual and auditory imprinting were tested independently in a sequential test, where novel and imprinted stimuli are presented in alternation. While other studies have used a simultaneous choice test (23), the sequential test allowed us to randomize stimulus presentation, measure baseline locomotion and assess the response to novel and imprinted stimuli independently.”

Regarding the reviewer’s concern about whether training for imprinting is sufficient to induce strong preference, we would like to argue that the response to the imprinted stimulus was substantially above baseline across trials. We enclose a figure from the previous rebuttal letter supporting this claim (Figure 7). The magnitude of imprinting induced by our training provided us a dynamic range to perform both loss- and gain-of-function manipulations.